# Expression of Interferon Regulatory Factor 8 (IRF8) and Its Association with Infections in Dialysis Patients

**DOI:** 10.3390/cells12141892

**Published:** 2023-07-19

**Authors:** Justa Friebus-Kardash, Fei Kuang, Tobias Peitz, Thamer A. Hamdan, Ute Eisenberger, Kristina Boss, Andreas Kribben, Karl Sebastian Lang, Michael Jahn

**Affiliations:** 1Department of Nephrology, University Hospital Essen, University of Duisburg-Essen, 47057 Essen, Germanymichael.jahn@krupp-krankenhaus.de (M.J.); 2Institute of Immunology, University Hospital Essen, University of Duisburg-Essen, 47057 Essen, Germany

**Keywords:** dendritic cells, dialysis, infections, interferon regulatory factor 8 (IRF8), monocytes

## Abstract

Patients on dialysis have dysfunctions of innate and adaptive immune system responses. The transcriptional factor IRF8 (interferon regulatory factor 8) is primarily expressed in plasmacytoid cells (pDCs) and myeloid dendritic cells (mDCs), playing a crucial role in the maturation of dendritic cells, monocytes, and macrophages, and contributing to protection against bacterial infections. The current study analyzed the expression patterns of IRF8 and assessed its association with the risk of infections in 79 dialysis patients compared to 44 healthy controls. Different subsets of leukocytes and the intracellular expression of IRF8 were measured using flow cytometry. Compared to the healthy controls, the dialysis patients showed significantly reduced numbers of pDCs and significantly increased numbers of natural killer cells and classical and intermediate monocytes. The dialysis patients exhibited decreased numbers of IRF8-positive dendritic cells (pDC *p* < 0.001, mDC1 *p* < 0.001, mDC2 *p* = 0.005) and increased numbers of IRF8-positive monocytes (*p* < 0.001). IRF8 expression in pDC, mDC, and classical monocytes was lower in the dialysis patients than in the controls. Dialysis patients who required hospitalization due to infections within one year of follow-up displayed significantly reduced IRF8 expression levels in pDCs compared to patients without such infections (*p* = 0.04). Our results suggest that reduced IRF8 expression in pDCs is a potential risk factor predisposing dialysis patients to serious infections.

## 1. Introduction

Despite the technical progress in recent years, mortality and morbidity rates continue to be high in patients with kidney failure with replacement therapy (KFRT) [1]. Infections are the second leading cause of death in patients, with mortality rates oscillating between 12% and 20% [2]. Abnormalities in the immune responses of the innate and adaptive immune systems are common in patients who undergo kidney replacement therapy (KRT), leading to an increased propensity for infections, especially bloodstream infections, and pneumonia [3,4,5]. The number of circulating immune cells and their functionalities are heavily impacted by kidney failure, which requires dialysis treatment and engenders a state of immunodeficiency in dialysis patients, contributing to susceptibility toward infections [6,7,8]. Plausible causes contributing to immune disorders in patients with KFRT include uremic toxins, malnutrition, increased oxidative stress accompanied by chronic inflammation, elevated apoptosis of immune cells, the choice of dialysis modality, and other dialysis-related effects such as dialysis membrane incompatibility, interactions between blood and dialysis equipment, presence of endotoxins in the water, high glucose concentrations in PD solutions, low pH, and the presence of glucose degradation products [9,10,11]. 

Interferon regulatory factor 8 (IRF8) plays a critical role in antimicrobial and antiviral defense [12]. IRF8 is a transcriptional factor that is predominantly expressed by dendritic cells and monocytes [12]. It consists of a DNA-binding domain and IRF association domain, which are required for cooperation with other transcriptional factors such as IRF1 and IRF2 [12,13]. The complex of IRF8 with IRF1 or IRF2 attaches to interferon-stimulated response elements, resulting in the activation of several genes, which are necessary for the maturation of the myeloid lineage and the consequent anti-infective immune response [13,14]. IRF8 drives the differentiation of myeloid cells in macrophages and facilitates the resistance of macrophages to intracellular pathogens [12,14,15,16]. There are plenty of data regarding severe immunodeficiency due to the loss of function mutations of IRF8 in humans [12]. In addition, mice lacking IRF8 were shown to have dramatically increased susceptibility to infections [17]. 

The current study was carried out to explore the expression of IFR8 in dendritic cells and monocytes in patients undergoing KRT in comparison to healthy subjects and to determine the relationship between IRF8 expression and the development of infection complications in a vulnerable group of dialysis patients. 

## 2. Materials and Methods

### 2.1. Study Populations 

The current single-center study included 79 adult patients with KFRT who were on the waiting list for transplantation or admitted for inpatient treatment at the University Hospital Essen. The study protocol was approved by the Ethics Committee of the University Hospital Essen, and all patients signed written informed consent (21-9883-BO). Hemodialysis was performed in 68 (86%) of 79 patients, and 11 (14%) of 79 patients were treated with peritoneal dialysis. Whole blood samples were collected from February to May 2021 at least 24 h after the last hemodialysis session. The clinical and laboratory data of the dialysis patients were obtained via a review of medical records and are listed in Table 1. 

The control population of healthy blood donors contained 44 adult coworkers without clinical signs of infection. The median age was 34 years, and 55% of the donors were women. Blood samples from the controls were obtained between February and May 2021.

All dialysis patients were free of clinical signs of infection at the time of analysis. In the second step of the prospective analysis, we contacted dialysis centers at the follow-up of one year to determine infection events requiring hospitalization, e.g., sepsis, pneumonia, or pyelonephritis. Among 79 study subjects, four patients underwent renal transplantation immediately after flow cytometry analysis and were excluded from the further prospective analysis. 

### 2.2. Flow Cytometry Analysis

Flow cytometry (FACS) analysis of whole blood samples was performed at the Institute of Immunology, University Hospital Essen. In order to prepare peripheral blood mononuclear cells (PBMCs), 10 mL of each whole blood sample was diluted with 10 mL of PBS. The PBMCs’ isolation was based on Ficoll–Paque PLUS density gradient centrifugation. For surface staining, cells were washed once in PBS and then incubated with primary monoclonal antibodies for 30 min at 4 °C. To determine the expression of IRF8 as a transcriptional factor in dendritic cells and monocytes, intracellular staining was carried out on PBMCs by using a fixation and permeabilization kit (Foxp3, BD Biosciences, Franklin Lakes, NJ, USA) according to the manufacturer’s instructions. The samples were acquired on a FACS Fortessa (BD, Franklin Lakes, NJ, USA) and analyzed via FlowJo software (Ashland, Katlitzburg, KY, USA). The gating strategy for the IRF8 expression in dendritic cells and monocytes is illustrated in Appendix A.

The following anti-human antibodies were used in the experiment set-up: Anti-Hu CD8a, Anti-Hu CD56, Anti-Hu CD123, Anti-Hu HLA-DR, Anti-Hu CD3, Anti-Hu/Mo IRF8 (1:100, eBioscience, Waltham, MA, USA), Anti-Hu CD16, Anti-Hu CD19 (1:100, BD Bioscience, Franklin Lakes, NJ, USA), Anti-Hu CD14, Anti-Hu CD141, Anti-Hu CD1c, Anti-Hu CD303, and Anti-Hu CD4 (1:100, Biolegend, San Diego, CA, USA).

### 2.3. Statistical Analysis

Categorical variables were expressed as numbers and percentages and were compared with the two-tailed χ^2^ test. Due to the non-normal distribution of continuous variables, data were given as median with interquartile range (IQR). To detect statistically significant differences between the two groups, the two-tailed Mann–Whitney test was used. Multivariable Cox regression analysis was used to verify whether the association between IRF8 expression in pDC and the appearance of infections requiring hospitalization upon one-year follow-up was independent of other covariates. To validate the predictive value of reduced IRF8 expression in pDC for infections requiring hospitalization, ROC analysis was performed. The level of statistical significance was set at *p* < 0.05. All data analyses were calculated with GraphPad Prism version 6 (GraphPad Software, Inc., La Jolla, CA, USA).

## 3. Results

### 3.1. Disturbed Numbers of Immune Cells in Patients with KFRT in Comparison to Healthy Individuals

Disorders of innate and adaptive immune systems are a well-known phenomenon in patients with chronic kidney diseases (CKD), particularly in those receiving kidney replacement therapies (KRT). Previous reports have described changes in the numbers and functionality of circulating immune cells in patients undergoing dialysis treatment [3,5,7].

In our study, we analyzed the distribution of cell types belonging to the innate and adaptive immune system in 79 dialysis patients and 44 healthy blood donors. Compared to healthy subjects, we observed a significantly reduced number of pDC in patients receiving KRT (Figure 1). On the contrary, the counts of natural killer (NK) cells, classical and intermediate monocytes, were higher among dialysis patients than among healthy blood donors (Figure 1). We found a decline of CD19-positive B cells in dialysis patients, whereas the numbers of circulating CD8^+^ and CD4^+^ T cells remained comparable in both groups (Figure 1).

### 3.2. Alterations in IRF8 Expression in Cells of Innate Immune System in Patients with KFRT

Next, we focused on differences in IRF8 expression in cells of the innate immune system between dialysis patients and healthy controls.

Dialysis patients displayed significantly elevated percentages of IRF8-positive mDC2 and monocytes containing all three subgroups (Figure 2). The numbers of IRF8-positive mDC1 and pDC were lower in dialysis patients than in the healthy population (Figure 2A). Strikingly, IRF8 expression was significantly reduced in mDC1, mDC2, and classical monocytes in patients on dialysis (Figure 2A). Considering IRF8 expression in pDC, we also recognized a trend toward decreased expression for dialysis patients (Figure 2A).

### 3.3. Patients Characteristics of Dialysis Patients and Healthy Controls

Our study cohort consisted of 79 patients with KRT. The median age was 56 years, and the portion of females was 44% (Table 1).

With regard to dialysis patients, the median dialysis vintage was 642 days (1.8 years) (Table 1). Fifteen out of 79 patients (19%) had undergone previous kidney transplantation (Table 1). The mean time period between transplantation and the start of KRT was 14 years, ranging from 2 to 34 years. Six out of fifteen patients (40%) received immunosuppression containing only prednisolone in a dosage of 5 mg. Another six patients (40%) were treated with calcineurin inhibitors in combination with prednisolone. Two patients (13%) received triple immunosuppressive therapy in reduced dosage, and one person was administered calcineurin inhibitor alone. The percentage of patients who were treated with peritoneal dialysis was 14% (Table 1). The most common cause of kidney failure in our study cohort was diabetic nephropathy (Table 1). The laboratory parameters of dialysis patients at the time point of FACS analysis are shown in Table 1. In the patients’ history before study inclusion, 29 out of 79 patients (37%) had experienced infections that required hospitalization, with sepsis occurring in 19 patients (24%) (Table 1). Within one year of follow-up, 20 out of 75 patients (27%) had infections requiring hospitalization, including 20 cases with septicemia and 8 with pneumonia (Table 1).

Due to the fact that the control group was significantly younger than the study cohort of dialysis patients, we supposed a potential age-related effect on changes in immune cell subsets demonstrated for dialysis patients. Therefore, we separated the study population of dialysis patients into old and young patients. Patients who were older than 65 years were counted as old patients. As shown in Appendix A, except for B cells and NK cells, which were reduced in older patients, the counts of immune cells were similar in both age groups (Appendix A). Likewise, the percentage of IRF8-positive dendritic cells and monocytes, and IRF8 expression in these cells, were not influenced by age (Appendix A).

Regarding the potential impact of the duration of KRT on counts of diverse immune cell subsets, we also correlated the duration of KRT with immune cell populations. As demonstrated in Appendix A, we found no correlation between the duration of KRT and different immune cell subsets or IRF8 expression in the cells of the innate immune system. Moreover, the three frequent causes of renal failure, such as diabetic nephropathy, nephrosclerosis, and chronic glomerulonephritis, as well as previous renal transplantations resulting in allograft failure and subsequent dialysis requirement, were not associated with different immune cell populations or IRF8 expression in certain immune cells (Appendix A).

### 3.4. Decrease of IRF8 Expression in pDC Was Associated with Higher Rates of Infections Requiring Hospitalization upon One-Year Follow-Up

Next, we questioned whether the above-mentioned reduction of IRF8 expression observed in our study cohort of dialysis patients might be linked to an increased rate of hospital admissions due to infections. Dialysis patients who developed infections requiring hospitalization within one year after the FACS analysis were characterized by elevated numbers of pDC, reduced expression of IRF8 in pDC, and slightly increased C-reactive protein (CRP) levels obtained parallel to the FACS analysis. Additionally, the proportion of previous transplantations was higher among patients with such infections than among those without (Table 2). While leukocyte counts did not differ, CRP levels were significantly elevated in patients hospitalized for infections within the follow-up year, which may indicate an aspect of chronic infection. However, rates of prior hospital admissions for infections did not differ between patients with and without infection-related hospitalizations within the follow-up year after FACS. This argues against the effect of chronic infection conditions, even though the effect cannot be fully elucidated (Table 2).

As illustrated by Figure 3, IRF8 expression was significantly decreased (*p* = 0.04) in patients who developed infections requiring hospitalization (Figure 3A). An area under the receiver-operating-characteristics curve (AUROC) of 0.66 shows a useful discriminative capacity for the occurrence of infections through IRF8 expression (Figure 3B). Dialysis patients with low IRF8 expression in pDC, lying below the median, tended to develop earlier and more frequent infections leading to hospital admissions during the one-year follow-up period (68% IRF8 expression below the median vs. 76% IRF8 expression above the median, Figure 3C). In a multivariable Cox regression analysis, reduced IRF8 expression in pDC exerted a minor effect on the risk of developing subsequent infections requiring hospitalization in a one-year follow-up (Table 2). In addition to the CRP value at the time point of inclusion, the history of previous renal transplantations appeared to be relevant for the prediction of infections qualifying for hospital admission in our cohort (Table 2).

## 4. Discussion

A comparison between dialysis patients and healthy controls revealed a significantly diminished number of pDC and B cells and significantly elevated counts of NK cells and classical and intermediate monocytes in patients receiving KRT. While the number of IRF8-positive pDC and mDC decreased, the IRF8-positive classical, intermediate, and non-classical monocytes were heightened in dialysis patients. The major finding of our study was the association of the reduction of IRF8 expression in pDC with a rising rate of infections requiring hospitalization occurring in the following year.

A large body of evidence proposes reduced numbers and impaired activity in polymorphonuclear leukocytes, hampering bactericidal responses in patients on dialysis [8,18]. Several reports, which are in line with our observations, mentioned dramatically decreased portions of pDC and mDC in patients with renal failure [9,19]. The impact of uremic toxins is a likely candidate for explaining the strong decline of dendritic cells by shortening their lifespan through the induction of apoptosis and necrosis [20]. Dialysis equipment, diminished release of GM-CSF, and increased turnover due to subsequent chronic inflammation in vessels might also negatively impact dendritic cell counts [9,19,21,22,23].

Abnormally increased proportions of monocyte populations in dialysis patients were also confirmed by numerous previous studies [5,8,18,24,25]. Dialysis patients are characterized by monocyte recruitment in the blood, raising the number of circulating monocytes in contrast to healthy subjects [8]. In concordance with our results, chronic kidney disease is associated, in particular, with augmentation of intermediate monocytes [25,26]. Uremia dampens phagocytic functions of monocytes and macrophages in patients with renal replacement therapy [6,20]. Other studies have reported increased expression of Toll-like receptors 2 and 4 on monocytes from dialysis patients, elevating their responsiveness to inflammatory stimuli [27].

In addition to the myeloid-derived immunosuppression in dialysis patients, a drop of B cells is a common phenomenon in patients on dialysis, which was also demonstrated in our cohort [28,29]. The uremic milieu mediates increased apoptosis and resistance to interleukin 17 and B cell activation factor (BAFF) of B cells [29]. Concerning the numbers of NK cells and T cells in patients receiving dialysis treatment, the results of previous reports are controversial [30,31]. Some previous studies shared our observation of elevated proportions of NK cells [30,32,33], while others reported decreased percentages or no changes [20,34,35]. Likewise, previous reports are conflicting regarding the numbers of circulating T cells under dialysis conditions [4,7]. While we found no changes in CD4^+^ and CD8^+^ T cells, studies on the immunomodulating effects of single hemodialysis sessions postulated an increased CD4^+^/CD8^+^ ratio as a consequence of CD8^+^ T cell decline [36]. Dialysis therapy affects the distribution of several T cell subsets with a deficiency in naïve T cells due to disrupted thymic output and accelerated aging of the T cell compartment [4,7,37].

To the best of our knowledge, this is the first study describing particularities of IRF8 patterns in dialysis patients and one of a few works addressing the relationship between alterations of immune cell numbers and the appearance of hospital admissions due to infections in dialysis patients [4,38,39]. We consider this approach particularly interesting because the immunomodulation of dialysis patients has not been fully elucidated. Uremia and dialysis treatment, causing an immunosuppressive state and changes in elements of the innate and adaptive immune systems of patients with KFRT, contribute to the increased prevalence of infectious complications in this sensitive patient group. On the other hand, IRF8-inducing differentiation of dendritic cells and type I interferon production by dendritic cells and monocytes play an essential role in the innate and adaptive immune response to infections. Vice versa, deficiency of IRF8 has been attributed to an elevated risk for infections in human and mice studies [12,17]. The loss of IRF8 expression due to mutations leads to primary immunodeficiency, which is characterized by life-threatening respiratory infections in childhood [12]. Considering the modulation of IRF8 expression seen in our study with dialysis patients, we speculate on potential reasons for this finding, such as modulation of IRF8 expression by the uremic environment or dialysis treatment. Probably, the numerical and functional alterations of dendritic cells might contribute to the reduced formation of IRF8. IRF8 requires ternary complexes with transcriptional factors and promotes the generation of type I interferon by myeloid cell lineages [12,40,41]. On the other hand, the activation of IRF8 occurs via interferon-mediated signaling through the IFNAR receptor, suggesting that IRF8 is a critical mediator of feedback amplification of the interferon response [40,41]. The previously described insufficient expression of superficial costimulatory molecules by dendritic cells and low number of pDC under dialysis therapy are likely to imply diminished interferon response [6]. Thus, the lack of type I interferon secretion might lower IFNAR-driven IRF8 induction, interrupting the feedback amplification loop.

The research group of Molony et al. published defective IRF8 expression in cells obtained from healthy adults who were older than 65 years [42]. Dialysis patients are marked by premature aging, which might also be involved in the observed reduction of IRF8 expression in dendritic cells. It is interesting to note that the comparison of IRF8 expression between young and old dialysis patients resulted in similar low IRF8 gMFI values in both groups.

As a main finding of the current work, we identified that dialysis patients with low levels of IRF8 expression in pDC developed more frequently and earlier serious infections during the one-year follow-up after the FACS analysis. The deficiency of transcription factor IRF8 is known to be associated with increased susceptibility to infections and reduced antiviral and antimicrobial defense mechanisms [12,40]. IRF8 partners with various interferon regulatory factors, exerting broad regulatory effects in the stimulation of myeloid cells and immune responses to infections [12,40,43]. In this way, IRF8 switches on numerous genes regulating recognition, processing, and presentation of antigens by myeloid cells and subsequent priming of T cells [12,43]. For example, the complex of IRF8 and IRF1, called IRF8/IRF1 regulome, activates macrophages via interferon-gamma, inducing essential antimicrobial defense signal pathways [40]. Therefore, IRF8 mutant macrophages were described to be susceptible to ex vivo infections with intracellular bacteria [15,16]. We anticipate that the reduction of IRF8 expression in pDC observed in our dialysis patients might impede the binding of IRF8 to other interferon-regulating factors or other transcriptional factors, leading to the deteriorated expression of instructive genes that are important for antigen presentation and T cell activation pathways. The deficit of IRF8 expression might be considered a potential risk factor triggering severe infections under dialysis conditions. Further studies are needed to understand the role of IRF8 in the complex and multifactorial dysregulation of the immune system in dialysis patients.

To our knowledge, reports on the role of other interferon regulatory factors besides IRF8 in the development of chronic kidney disease are rare, and reports on potential variations in the expression of other IRFs under dialysis conditions are lacking. The research group of Li et al. described IRF1’s involvement in the pathogenesis of renal fibrosis, enabling progression toward chronic kidney disease [44]. Patients with chronic kidney disease and histological evidence of fibrotic processes in their kidneys displayed elevated levels of IRF1 [44]. Mice that underwent unilateral obstruction or adriamycin application also had an augmentation of IRF1 expression in tubular epithelial cells [44]. On the other hand, the depletion of IRF1 in mice restricted the progression of renal fibrosis. IRF1 was shown to act via the suppression of Klotho by downregulation of the promotor of the Klotho gene, C/EBP- beta [44]. Another work by Lorenz et al. addressed the negative effect of IRF4 on the progression of ischemia-reperfusion injury toward chronic kidney disease with renal fibrosis in a mouse model [45]. IRF4 mediates tissue remodeling and promotes polarization of macrophages and T-cell differentiation [45]. Mice with IRF4 deficiency developed chronic kidney disease [45]. Furthermore, in a murine model of ischemia-reperfusion injury proceeding toward chronic kidney disease, an increased expression of IRF4 was detected, suggesting the regulatory role of IRF4 in the chronic phase [45]. IRF4-deficient mice exhibited defects in macrophage polarization, leading to the infiltration of macrophages and the release of pro-inflammatory cytokines by macrophages, subsequently causing intrarenal chronic inflammation and renal fibrosis [45]. However, the data on the relationship between IRF1 or IRF4 and chronic kidney disease mainly come from murine studies, and conclusions on the clinical significance of IRF1 and IRF4 for chronic kidney disease should be elucidated in future studies.

However, we are aware of several limitations of the present study. Notably, our healthy control population consisted of subjects who were significantly younger than the study collective of dialysis patients. Subsequent subdivision of our dialysis patient cohort into patients older and younger than 65 years showed no significant differences among different immune cell populations, except for an age-dependent reduction of NK cells and B cells (Appendix A), making age a conceivable factor impacting cells of the adaptive immune system. Nevertheless, we cannot exclude the effects of dialysis-related factors on immune cell counts and IRF8 expression in the cells of the innate immune system. Unfortunately, we did not conduct a comparison between patients with end-stage chronic kidney disease without KRT and dialysis patients in our study. Hence, conclusions on disorders of immune cell populations and IRF8 expression in patients with chronic kidney disease without KRT cannot be fully extrapolated from our results derived from patients with KFRT. The usage of immunosuppressive treatment in low dosage in 15 out of 79 patients due to previous renal transplantations may affect the counts of different immune cell populations and IRF8 expression on dendritic cells and monocytes, which represents another limitation of the current study. Repeated measurements of IRF8 could provide a better estimation of potential changes and dynamics of the specific immune cells involved, which were not covered by our one-off measurement. Furthermore, the number of serious infection events was limited and could potentially be increased by prolonging the follow-up time of observation and including more dialysis patients. Studies with larger cohorts are required to determine whether IRF8-positive cell proportions or IRF8 expression on different immune cells could serve as suitable parameters for predicting infection events. Additionally, the inter-individual dynamics of IRF8 expression and the effects of dialysis-specific interventions on it need to be further investigated.

## 5. Conclusions

The reduction of IRF8 expression in plasmacytoid dendritic cells could be associated with the development of serious infections in dialysis patients during the one-year follow-up. Our data, which demonstrate that dialysis patients with low IRF8 expression are prone to severe infections, support the central role of IRF8 in protecting against infections.

## Figures and Tables

**Figure 1 cells-12-01892-f001:**
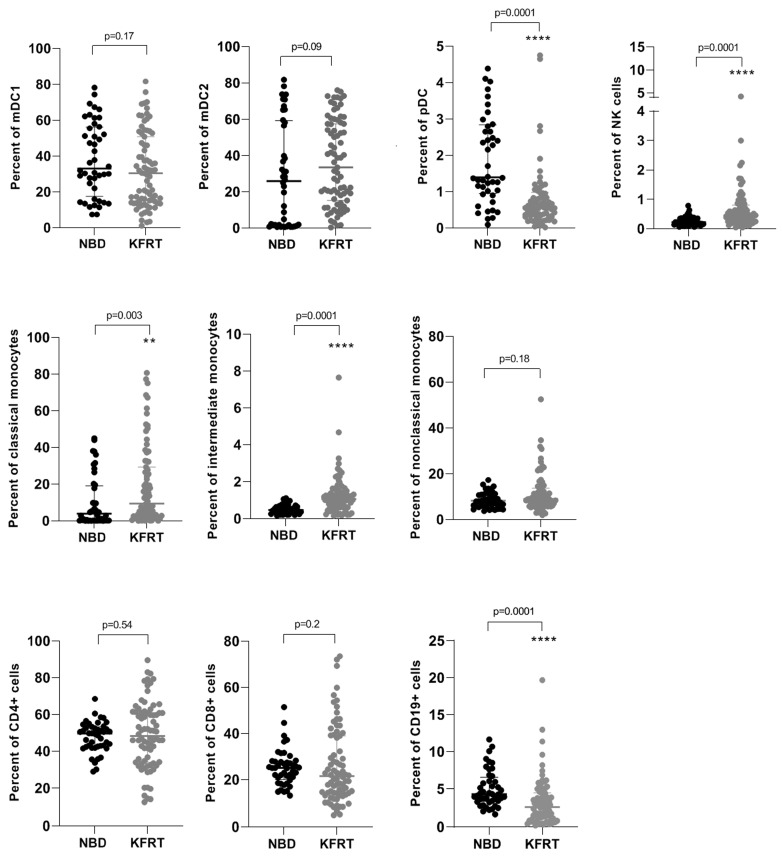
Changes in cell numbers of immune cell subsets comparing 79 dialysis patients with 44 healthy controls. **—*p* = 0.01; ****—*p* ≤ 0.0001; KFRT; kidney failure receiving replacement therapy; mDC, myeloid dendritic cells; NBD, normal blood donors; NK, natural killer; pDC, plasmacytoid dendritic cells.

**Figure 2 cells-12-01892-f002:**
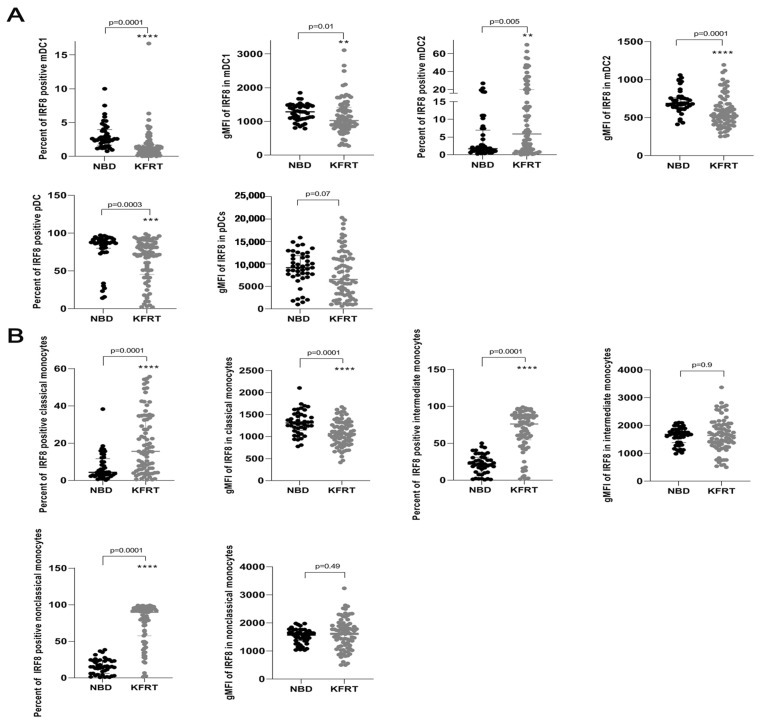
Changes in IRF8 expression in dendritic cells and monocytes comparing 79 dialysis patients with 44 healthy controls. (**A**) Differences in counts of IRF8-positive dendritic cells and IRF8 expression in each dendritic cell between dialysis patients and control subjects. (**B**) Differences in counts of IRF8-positive monocytes and IRF8 expression in each monocyte between dialysis patients and control subjects. **—*p* = 0.01; ***—*p* = 0.001; ****—*p* ≤ 0.0001; KFRT; kidney failure receiving replacement therapy; gMFI, geometric mean; IRF8, interferon regulatory factor 8; mDC, myeloid dendritic cells; NBD, normal blood donors; pDC, plasmacytoid dendritic cells.

**Figure 3 cells-12-01892-f003:**
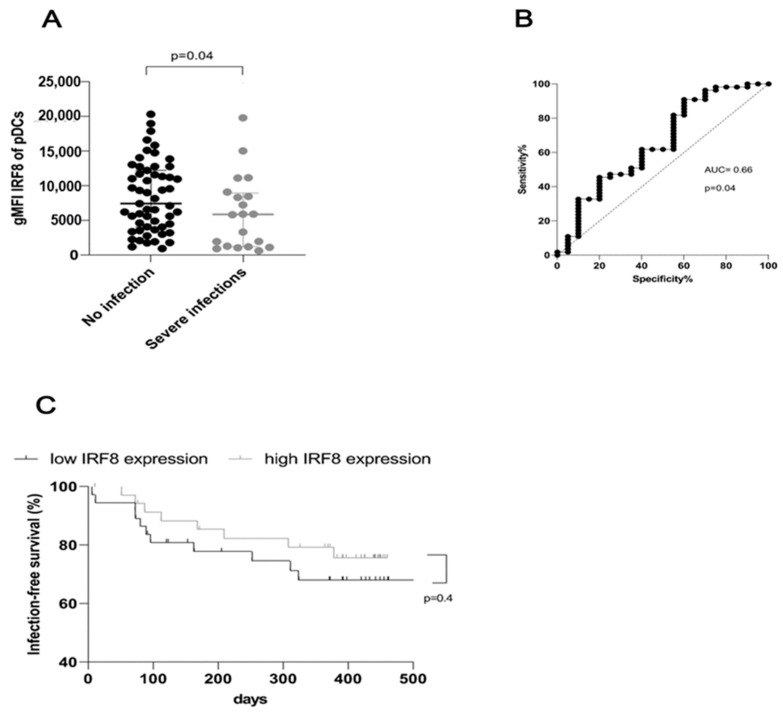
Impaired IFR8 expression in plasmacytoid dendritic cells is associated with an increased rate of infections requiring hospital admissions upon follow-up of one year after the FACS analysis. (**A**) Dialysis patients who experienced hospital treatment of infections within one year after the FACS measurement displayed lower IRF8 expression in plasmacytoid dendritic cells than patients without such episodes at the one-year follow-up (*p* = 0.04). (**B**) ROC analysis of IRF8 expression in plasmacytoid dendritic cells as a potential predictor for upcoming severe infections (AUC = 0.7, *p* = 0.04). (**C**) Infection-free survival dependent on IRF8 expression in plasmacytoid dendritic cells after dividing 75 dialysis patients into two equal groups according to the median IRF8 expression (*p* = 0.41).

**Table 1 cells-12-01892-t001:** Demographical and laboratory characteristics of 79 study patients with kidney failure receiving replacement therapy (KFRT).

**Demographic characteristics**	
Age, median (range)	56 (26–86)
Number of women, n (%)	35 (44)
Previous renal transplant, n (%)	15 (19)
Dialysis vintage (days), median (range)	642 (5–5476)
Peritoneal dialysis patients, n (%)	11 (14)
**Laboratory parameters**	
Creatinine (mg/dL, 65/79), mean (range)	7.4 (2.3–15.2)
Urea nitrogen (mg/dL, 62/79), mean (range)	52.4 (8.3–106.7)
Parathyroid hormone (pg/mL, 42/79), mean (range)	372 (2.8–1100)
Vitamin D3 (ng/L, 28/79), mean (range)	29.1 (10.6–58.6)
Serum albumin (g/dL,50/79), mean (range)	3.6 (1.1–5.2)
Total serum protein (g/dL, 56/79), mean (range)	6.1 (4.4–7.8)
Hemoglobin (g/dL, 50/79), mean (range)	10.2 (6.3–14.1)
C-reactive protein (mg/dL, 47/79), mean (range)	3.2 (0.4–13.7)
Leukocytes (/nL, 60/79), mean (range)	7.2 (1.9–25.8)
**Causes of renal failure**	
Diabetic glomerulosclerosis, n (%)	16 (20)
Chronic glomerulonephritis, n (%)	14 (18)
Nephrosclerosis, n (%)	8 (10)
Polycystic kidney disease, n (%)	5 (6)
Tubulointerstitial nephritis, n (%)	4 (5)
Congenital anomalies, n (%)	6 (8)
Autoimmune disease, n (%)	3 (4)
Reflux nephropathy/recurrent pyelonephritis, n (%)	2 (3)
Other, n (%)	21 (27)
**Previous infections**	
Infections requiring hospitalization, n (%)	29 (37)
Sepsis, n (%)	19 (24)
Pneumonia, n (%)	19 (24)
Pyelonephritis, n (%)	5 (6)
**Infections within one year after analysis (75 patients)**	
Infections requiring hospitalization, n (%)	20 (27)
Sepsis, n (%)	15 (20)
Pneumonia, n (%)	8 (11)
Pyelonephritis, n (%)	4 (5)

**Table 2 cells-12-01892-t002:** Results of univariate and multivariate analysis identifying risk factors for the development of infections requiring hospitalization upon a follow-up period of one year after the initial FACS analysis among 75 dialysis patients.

Variable	Infection Requiring Hospitalization Median (IQR) or n (%)	No Infection or Infection Not Requiring Hospitalization Median (IQR) or n (%)	χ^2^ Square	OR	*p* Value	Multivariate Relative Risk (95% CI)	*p* Value
Patients	20	55			NA		
mDC1	30.0 (14–53)	26.2 (13.9–46.2)			0.95		
mDC2	21.4 (15.5–60.4)	38.9 (18.1–57.5)			0.60		
pDC	0.7 (0.5–1.8)	0.6 (0.3–0.8)			0.02	1.05 (0.88–1.26)	0.58
NK cells	0.5 (0.3–0.7)	0.5 (0.3–0.9)			0.48		
Classical monocytes	4.1 (2.9–27.1)	12.5 (3.5–32.3)			0.39		
Intermediate monocytes	1.1 (0.6–1.6)	1.0 (0.6–1.5)			0.86		
Non-classical monocytes	9.0 (5.4–14)	8.8 (6.5–11.8)			0.78		
CD4^+^ cells	47.8 (32–64.6)	49.1 (36.6–61.0)			0.93		
CD8^+^ cells	21.2 (13.9–40.6)	20.4 (14–31.0)			0.83		
CD19^+^ cells	2.5 (0.8–4.2)	2.6 (1.3–4.7)			0.80		
% IRF8 positive mDC1	1.1 (0.5–1.4)	1.1 (0.5–1.9)			0.61		
gMFI of IRF8 in mDC1	985 (713–1402)	1033 (838–1417)			0.46		
% IRF8 positive mDC2	4.9 (1.3–13.5)	6.3 (1.6–25.5)			0.64		
gMFI of IRF8 in mDC2	524 (405–654)	540 (417–738)			0.63		
% IRF8 positive pDC	71.0 (11.9–81.2)	75.6 (51.4–88.2)			0.12		
gMFI of IRF8 in pDC	5862 (1216–8942)	7421 (4055–12,223)			0.04	1 (1.0–1.0)	0.34
% IRF8 positive class. monocytes	15 (3.7–27.3)	15.8 (6.0–34.8)			0.35		
gMFI of IRF8 in class. monocytes	1070 (768–1230)	1046 (855–1317)			0.41		
% IRF8 positive interm. monocytes	71.1 (22.5–87.2)	76.2 (51.4–87.8)			0.48		
gMFI of IRF8 in interm. monocytes	1662 (855–2020)	1640 (1288–2026)			0.50		
% IRF8 positive non-class. monocytes	79.1 (32.1–94.5)	90.5 (59.7–95.3)			0.27		
gMFI of IRF8 in non-class. monocytes	1458 (897–1884)	1618 (1179–1926)			0.31		
Age (years)	59 (46–69)	54 (44–63)			0.34		
Female gender	10 (50%)	23 (46%)	0.40	1.39	0.53		
Previous renal transplants	7 (35%)	8 (22%)	3.84	3.16	0.05	7.78 (2.02–30.0)	0.003
Dialysis vintage (days)	553 (46–1541)	664 (361–1825)			0.32		
Peritoneal dialysis patients	1 (5%)	10 (18%)	2.04	0.24	0.15		
Creatinine (mg/dL)	6.9 (3.8–9.2)	6.5 (4.8–9.6)			0.40		
Urea nitrogen (mg/dL)	54 (38–69)	49 (37–63)			0.59		
Parathyroid hormone (pg/mL)	187 (60–517)	340 (207–637)			0.22		
Vitamin D3 (ng/L)	22 (12–36)	31 (22–38)			0.17		
Serum albumin (g/dL)	3.5 (3.0–4.3)	3.8 (3.1–4.2)			0.67		
Total serum protein (g/dL)	5.9 (5.2–6.8)	6.3 (5.5–6.8)			0.55		
Hemoglobin (g/dL)	10.5 (8.8–11.5)	10.4 (8.3–11.9)			0.93		
Leukocytes (/nL)	6.7 (4.7–9.0)	6.8 (4.9–8.6)			0.95		
C-reactive protein (mg/dL)	3.9 (1.4–7.1)	1.8 (0.5–4.3)			0.04	1.45 (1.16–1.81)	0.001
Previous infections	10 (50%)	18 (33%)	1.90	2.10	0.17		
Previous sepsis	5 (25%)	14 (26%)	0.00	0.98	0.97		
Previous pneumonia	4 (20%)	14 (26%)	0.24	0.73	0.63		
Previous pyelonephritis	3 (15%)	2 (4%)	3.04	4.68	0.08		

IQR—interquartile range; IRF8—interferon regulatory factor 8; mDC—myeloid dendritic cells; NK—natural killer; OR—odds ratio; pDC—plasmacytoid dendritic cells, class. monocytes—classical monocytes; interm. monocytes—intermediate monocytes, non-class. monocytes—non-classical monocytes.

## Data Availability

The data that support the findings of this study are available from the corresponding author upon reasonable request.

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
