# Peer review of "Expression of Interferon Regulatory Factor 8 (IRF8) and Its Association with Infections in Dialysis Patients"

_cells, 2023, doi:10.3390/cells12141892_

Round 1
Reviewer 1 Report
In this study, the authors have demonstrated that the expression of IRF8 in selected innate immune cells is lower in dialysis patients compared to controls. Furthermore, they have identified the reduced IRF8 expression in pDC as a potential risk factor predisposing dialysis patients to serious infections. The data collected from patients in this manuscript are valuable, there are minor issues that should be addressed by the authors:
Considering that members of the IRF family contribute significantly to the immune response against infections, why did the authors specifically select IRF8 for their study instead of other IRFs? Moreover, do the expressions of other IRFs also undergo changes in dialysis patients?
Reviewer 2 Report
The authors examined IFR8 expression in dendritic cells and monocytes of patients receiving kidney replacement therapy (KRT) compared to healthy controls to determine the mechanisms of the immunocompromised state in dialysis patients and the role of IFR8 in this state.
This is an interesting study that may shed light on an important mechanism of immunocompromise in CKD patients. However, there are some concerns, including the following:
1. The authors clearly demonstrated that the number of IRF8-positive immune cells is reduced in KRT patients and is associated with an increased incidence of serious infections. In other words, they showed that IRF8 reflects the immunodeficient state of KRT patients. However, the finding of reduced immunocompetence and a concomitant increased incidence of infections in KRT patients compared to healthy controls is not so surprising; more data and discussion on the mechanism of reduced IRF8-positive immune cells in KRT patients would be helpful.
Examples:
The authors confirm that there are no age-related differences in the number of immune cells, the percentage of IRF8-positive dendritic cells and monocytes, and the expression of IRF8 in these cells. However, these may be influenced by the duration of KRT (5-5476 days). Have the authors examined the relationship between KRT duration and immune cell populations?
Similarly, were there any relationships between the causes of renal failure and the immune cell population, including IRF8-positive cells?
Also, could renal replacement therapy itself have affected the number of immune cells in the blood? The study did not include CKD patients not receiving KRT, so data may not be available, but the authors could add comments in the discussion.
2. Univariate and multivariate analysis of risk factors for the development of severe infections during the 1-year follow-up period after the initial FACS analysis (Table 2) seems to indicate elevated CRP as one of the characteristics. Is it possible that patients with chronic infectious diseases were included in the severe infection group?  It would be beneficial to add white blood cell counts to Tables 1 and 2.
In this connection, the definition of severe infection should also be reconsidered. In this study, all patients with infections that required hospitalization were defined as severe cases, which may be influenced by the age of the patient; i.e., older patients may be more likely to be hospitalized for the same infection.
Minor points:
1. Figure 1 shows a significant increase in the number of macrophages, but not monocytes, in the peripheral blood of patients undergoing KRT. A certain percentage of macrophages are also detected in healthy individuals, but they are less common in clinical practice. What markers are used to identify these cells as macrophages? 
2. Please clearly state the definition of a patient who has had a renal transplant, especially if immunosuppressants were still being used at the time of the study. 
